# Two-Dose Vaccination Significantly Prolongs the Duration from Symptom Onset to Death: A Retrospective Study Based on 173,894 SARS-CoV-2 Cases in Khyber Pakhtunkhwa, Pakistan

**DOI:** 10.3390/ijerph191811531

**Published:** 2022-09-13

**Authors:** Qianqian Song, Naseem Asghar, Ata Ullah, Baosheng Liang, Mengping Long, Taobo Hu, Xiaohua Zhou

**Affiliations:** 1Department of Biostatistics, Peking University, Beijing 100191, China; 2Department of Statistics, Abdul Wali Khan University, Mardan 23200, Pakistan; 3Division of Life Science, Hong Kong University of Science and Technology, Clear Water Bay, Kowloon, Hong Kong, China; 4Department of Pathology, Peking University Cancer Hospital, Beijing 100083, China; 5Chongqing Research Institute of Big Data, Peking University, Chongqing 401121, China; 6Department of Breast Surgery, Peking University People’s Hospital, Beijing 100044, China

**Keywords:** COVID-19, symptom onset to recovery, symptom onset to death, case fatality rate, vaccine dose

## Abstract

This research was carried out to quantify the duration from symptom onset to recovery/death (SOR/SOD) during the first four waves and the Alpha/Delta period of the epidemic in Khyber Pakhtunkhwa, Pakistan, and identify the associated factors. A total of 173,894 COVID-19 cases were admitted between 16 March 2020 and 30 November 2021, including 458 intensive care unit (ICU) cases. The results showed that the case fatality rate (CFR) increased with age, and females had a higher CFR. The median SOR of ICU cases was longer than that of non-ICU cases (27.6 vs. 17.0 days), while the median SOD was much shorter (6.9 vs. 8.4 days). The SOR and SOD in the Delta period were slightly shortened than the Alpha period. Age, cardiovascular diseases, chronic lung disease, diabetes, fever, breathing issues, and ICU admission were risk factors that were significantly associated with SOD (*p* < 0.001). A control measure, in-home quarantine, was found to be significantly associated with longer SOD (odds ratio = 9.49, *p* < 0.001). Infected vaccinated individuals had longer SOD than unvaccinated individuals, especially for cases that had received two vaccine doses (*p* < 0.001). Finally, an advice on getting full-dose vaccination is given specifically to individuals aged 20–59 years.

## 1. Introduction

On 26 February 2020, the first coronavirus disease 2019 (COVID-19) case was confirmed in Sindh province. This case had traveled to Iran and returned to Pakistan by air on 20 February 2020 [1]. By 16 March 2020, 15 cases had been confirmed in the Khyber Pakhtunkhwa (KP) province [2]. To date, KP is one of the provinces with the highest number of confirmed cases, with the highest case fatality rate (CFR). By 30 November 2021, KP had 180,075 confirmed cases, 5846 deaths, and 173,495 recoveries, and it had roughly experienced four waves of COVID-19 (See Figure 1) [2]. In addition, several variants of SARS-CoV-2 emerged in that period, notably Alpha, Beta, Gamma, and Delta. The Alpha variant was first reported in Pakistan on 29 December 2020. It triggered a surge in cases in March 2021 [3] and dominated the total case numbers (82.5%) confirmed in April [4]. The Beta variant was first reported in the capital Islamabad around April 2021, and cases then rapidly increased in May and June [4]. On 3 May 2021, the Delta variant was also found [5]. No cases of the Beta variant were reported from 7 July 2021 onwards [6]. The prevalence of the Delta variant coincided with the increase in number of hospitalizations and CFR.

There have been a few studies discussing the epidemiological and clinical characteristics of patients and a few studies on vaccine doses and outcomes in Pakistan. The results showed that males had a higher incidence of disease and higher mortality rate [7] and that strict preventive measures for high-risk groups were essential [8]. Comorbidity, oxygen saturation, dyspnea on arrival, and length of hospital stay were also associated with a higher case fatality rate [9]. To our knowledge, in the existing literature, the sample size used to study the epidemiological and clinical characteristics of COVID-19 was small, and most studies were related to early stages of the epidemic. Currently, there are few studies that have estimated the distribution of symptom onset to recovery (SOR)/death (SOD), focused on the effect of basic characteristics on the length of SOR and SOD, or studied the importance of vaccination.

As is known to all, vaccination against COVID-19 is considered to be an effective way to control the pandemic, and its effectiveness depends heavily on the overall vaccination coverage. On 2 February 2021, Pakistan started its vaccination rollout. By 30 November 2021, the number of people vaccinated per hundred exceeded 53.17%, and the number of people fully vaccinated per hundred was over 21.69% [10]. From 2 February 2021 to 30 November 2021, different types of COVID-19 vaccines were approved for distribution in Pakistan, including CanSino, Sinopharm/Beijing, and Sputnik V, followed by Sinovac, Oxford/AstraZeneca, Covaxin, Moderna, and other vaccine brands such as Pfizer/BioNTech [10]. The government of Pakistan was providing CanSino, Sinopharm, Sinovac, and AstraZeneca vaccines free of cost. Health care workers were the first to be vaccinated [11]. After 16 March 2021, vaccination for different age groups began [12]. Lack of understanding and awareness of risk, safety, formal education, and low household income were some of the reasons for low vaccination rates in Pakistan [13,14]. Vaccination studies in other countries or regions have shown that vaccination could reduce the risk of transmission and hospitalization [15,16,17,18] and that an increase in vaccine dose (two or three doses) could significantly reduce the risk of hospitalization [19,20] and symptomatic and asymptomatic infections [21]. However, there have limited studies so far investigating the effect of vaccine doses on SOR or SOD in Pakistan based on real-world evidence.

As the outbreak progresses in Pakistan, reviewing epidemiological characteristics, estimating epidemiological parameters, and analyzing the effect of vaccines on outcomes could help increase people’s acceptance of vaccines and improve vaccination rates. We note that determining the length of SOR and SOD can allow us predict the infectivity of the virus so that residents can realize the importance of seeking treatment after symptom onset and further reduce the proportion of under-reported cases.

In this study, we examined the epidemiological characteristics of COVID-19 cases from 16 March 2020 to 30 November 2021 and estimated the distribution of SOR and SOD, especially for cases during the Alpha and Delta periods. Then, we evaluated factors associated with the length of SOR and SOD. In particular, SOR and SOD of individuals who were infected even after vaccination were investigated to explore their association with vaccine dose.

## 2. Data and Methods

### 2.1. Data Sources

The dataset was acquired from the office of the Directorate General Health Services (KP), Pakistan. This office is responsible for the health system at all levels based on the primary health care approach through the district health system aimed at ensuring universal access to quality health care in the KP province. The dataset includes all hospital attendance records reported from all districts of the KP province to the Directorate during the four waves of SARS-CoV-2 infection. We approached the office of the Additional Director General Health, Khyber Pakhtunkhwa, for permission to study the provincial SARS-CoV-2 data available with the department.

Data from 16 March 2020 to 30 November 2021 were collected, with a total of 173,894 confirmed cases, among which 458 were intensive care unit (ICU) cases. The date of symptom onset was defined as the day when symptoms first emerged. Comorbidities were based on the patient’s self-reported history and clinical records in the epidemiologic survey. Vaccine doses included 0, 1, and 2, in addition to cases whose vaccination status was not recorded. Patient status included active, dead, and recovered, where active means cases were still alive in the hospital at the last observation.

According to official media reports and the trend of weekly new cases in Figure 1, we define the range of the first wave as being from 16 March 2020 to early August 2020 and the second wave from early November 2020 to mid-February 2021. The third wave started in mid-March 2021 and ended in late June 2021 [22], and the fourth wave started in early July 2021 [23] and faded in late October 2021. In addition, the periods of Alpha and Delta infection in KP were roughly from 26 February 2021 to 15 May 2021 and 7 July 2020 to 30 November 2021, respectively (see Figure 1).

Figure 2 gives the flowchart of data selection. We first removed cases with missing data on age, gender, and date of symptom onset. Cases with recorded recovery or death dates that were earlier than the symptom onset date were also excluded from our analysis.

### 2.2. Statistical Analysis

Categorical variables were summarized using frequencies (percentages), and continuous variables were described by medians (interquartile range, IQR). The chi-square test and Fisher’s exact test were used to draw comparisons for discrete variables, and the Kruskal–Wallis rank-sum test was used for continuous variables.

The case fatality rate (CFR) was defined as the proportion of deaths among all confirmed cases over a certain period [24,25]. We chose the duration from symptom onset to recovery (SOR) and the duration from symptom onset to death (SOD) as two outcomes of interest. Because there were 1014 cases where the recorded dates of symptom onset and death or recovery were on the same day, thereby resulting in the observed duration (in days) to be zero, we let the duration of these cases be 0.5 days to reduce bias [26].

We tried four parametric survival models, namely, Weibull, Gamma, log-normal, and log-logistic models, to fit the distributions of SOR and SOD. The best model was selected using Akaike’s information criterion (AIC). The 95% confidence intervals (CIs) of model parameters were constructed using the bootstrap method with 1000 resampling. We employed accelerated failure time (AFT) models to evaluate factors associated with SOR and SOD and quantified the effects using the odds ratio (OR)/hazard ratio (HR) [27,28,29] and their 95% CIs. We included age, gender, symptoms (fever, sore throat, cough, diarrhea, breathing issue, and headache), basic comorbidities (cardiovascular disease including hypertension, chronic lung disease, diabetes, and obesity), home quarantine status, ICU admission, and ventilator status in the AFT models. The Kaplan–Meier method was used to compare the survival probability of cases with or without vaccination.

Through propensity score matching (PSM), we matched clean data 1:1 by gender and age during the Alpha and Delta periods to compare their epidemiological parameters. In addition, through multiple PSM, we matched cases (1:1) with recorded vaccine doses (zero, one, or two) by all characteristics except for vaccine doses in all periods, the fourth wave, and the Delta period to quantify the odds ratio of vaccine doses on SOR and SOD. All analyses were performed using R (version 4.2.0) with the following packages: fitdistrplus, flexsurv, forestmodel, Matching, and ggplot2. A *p*-value smaller than 0.05 was considered statistically significant.

## 3. Results

### 3.1. Epidemiological Characteristics of COVID-19 Cases

Table 1 presents the characteristics of all confirmed cases. Overall, 63.1% of cases were male, with a median age of 36 years. ICU cases were much older (60 vs. 36) and had a higher proportion of males than non-ICU cases (74.7% vs. 63.0%). From the descriptive analysis, 69.9% of cases were between the ages of 20 and 59. A total of 5.2% cases had been vaccinated (3.0% had two doses, 2.2% had one dose). It is worth noting that all ICU cases were reported before the vaccination rollout date (2 February 2021 [10]). Many cases had one or more symptoms or basic diseases. The most common symptoms were fever and cough, while cardiovascular diseases including hypertension and diabetes were the two most prevalent basic diseases. Compared to non-ICU cases, 79.3% of cases were put on ventilators after ICU admission, and the mortality rate was as high as 87.3%. The ICU cases had a much lower proportion of home quarantine than the non-ICU cases (0.2% vs. 59.0%).

Through PSM, 38,653 cases were matched to the Alpha and Delta periods. In the Delta period, 4986 cases had received two vaccine doses and 3739 cases had received one vaccine dose, which is significantly more than in the Alpha period. Compared to the Alpha period, the Delta period had an increase in the proportion of deaths and patients still in hospital and a decrease in the proportion of recovery. Fever and cough and cardiovascular disease including hypertension and diabetes remained the two symptoms and underlying diseases, respectively, with the highest proportions. The proportion of cases in home quarantine during the Delta period was 51%, which was 13.8% lower than that in the Alpha period.

Figure 3 depicts the CFR stratified by gender and age in the four waves, subgroups, and the Alpha and Delta periods. As shown in Figure 3a, the second wave had the lowest CFR, and the CFR of females was higher than males during the whole epidemic except for the second wave. As shown in Figure 3c, females had larger CFR than males in all subgroups, and the rate of males over females in the ICU was approximately 1. As shown in Figure 3e, the CFR during the Alpha and Delta periods had a similar trend, with females having slightly higher CFR than males. The overall CFR of Delta was slightly higher than that of Alpha. The CFR gradually increased with age, with the highest CFR in the age group over 80 years (Figure 3b,d,f).

### 3.2. Survival Models and Median Estimates of SOR/SOD

#### 3.2.1. Estimates of SOR/SOD

Table 2 gives the median estimates and 95% CIs of SOR/SOD. The log-logistic distributions provided the best fit for SOR and SOD. The estimated median of SOR was 27.6 days for ICU cases and up to 247.0 days, while the median was 17.0 days for non-ICU cases and the longest was just 56.1 days. The SOD of ICU cases was 1.5 days shorter than that of non-ICU cases (6.9 vs. 8.4 days). Cases during the Delta period had shorter SOR and SOD than that during the Alpha period (15.0 vs. 16.0 days, 9.2 vs. 9.7 days). Figure 4 showed that the best-fitted distribution of SOR and SOD were left-skewed in different subgroups, and both SOD and SOR had long right tails.

#### 3.2.2. Estimates of SOR/SOD Grouped by Vaccine Doses

Table 3 presents the medians of SOR and SOD stratified by vaccine doses among cases with recorded vaccine information. The results showed that the length of SOR and SOD increased with the number of vaccine doses. The SOR and SOD of cases with two vaccine doses were 16.5 and 11.2 days, respectively, much longer than other cases. We noted that unvaccinated cases had the shortest SOD with a median of 9.1 days.

#### 3.2.3. Estimates of SOR/SOD in the Four Waves

Table 4 presents the estimated medians of SOR and SOD in the four waves. The results showed that the median of SOR in the first wave was the longest, but the SOD in the first wave was the shortest. In the second wave, where the number of confirmed cases was the lowest (see Figure 2), the median SOR was shortened by 8.6 days and the SOD was extended by 1.5 days. During the third and fourth waves, the SOR and SOD were both shortened.

### 3.3. Factors Associated with SOR and SOD

Table 5 presents the adjusted ORs using the AFT models for SOR and SOD. A longer SOR was observed in males and older cases. The older the case, the longer was the SOR (OR > 1, *p* < 0.001).

Cases with symptoms had a longer SOR than those who did not have such symptoms, except for cases with headache (OR = 0.98, *p* = 0.009). We noted that cases with ventilators had a 3.82-fold longer SOR than cases without ventilators. With the development of the epidemic, the SOR was significantly shortened. No significant differences in SOR were identified among cases with basic diseases.

As the results in Table 5 indicate, males had a longer SOD than females (OR = 1.13, *p* < 0.001). Compared to cases under 20 years old, the SOD decreased with increasing age, that is, the older the case, the shorter was the SOD (OR 1, *p* < 0.001). We noticed that once cases developed fever and breathing issues, their SOD was shortened significantly (OR = 0.66, 0.19, *p* < 0.001); however, sore throat and headache were significantly associated with a longer SOD (OR = 1.13, 1.32, *p* < 0.001). Cases with cardiovascular diseases including hypertension, chronic lung disease, and diabetes had a shorter SOD than cases without these diseases (OR < 1, *p* < 0.001), while obesity could significantly prolong the SOD (OR = 4.10, *p* < 0.001). Interestingly, in-home quarantine could greatly prolong SOD (OR = 9.49, *p* < 0.001). However, once admitted to ICU and put on a ventilator, the SOD was greatly shortened (*p* < 0.001). Compared to cases diagnosed from 16 March 2020 to 18 September 2020, cases diagnosed after 15 February 2021 had shorter SOD. No significant differences in SOD were found among cases with cough and diarrhea.

To assess the effect of the number of vaccine doses on SOR and SOD during the fourth wave and Delta variant, we performed multiple PSM for all characteristics except vaccine doses for cases (see Figure 5a). Figure 5b presents the results of the AFT analysis. The results showed that full (two) dose was more effective than partial or no dose in prolonging SOD (OR = 2.11, 3.82, *p* < 0.001). The overall survival time of unvaccinated cases was shorter than the survival times of vaccinated cases. The result of SOR displayed a minor difference between the effects of two doses and one dose on SOR (data not shown). On the contrary, the SOD of cases with two doses was much longer than that of cases with one dose (see Figure 5b). Compared to unvaccinated cases, the SOD was up to 9.21-fold in cases with two vaccine doses, and the maximum difference from OR of one dose was 3.48.

As shown in Figure 6, the survival probability of infected cases with two vaccine doses was higher than that of cases with one dose, and the survival probability of infected cases after vaccination was significantly higher than that of unvaccinated cases. The survival probability of vaccinated cases decreased the most during 0–25 days and then was almost unchanged. We noted that the survival probability of a single dose was lower than that of two doses. However, the survival probability of infected cases without vaccination dropped sharply from 0 to 50 days and was lower than that of vaccinated cases.

## 4. Discussion

The obtained results showed that cardiovascular diseases and diabetes were the two most prevalent basic diseases among the confirmed cases. In addition, males had a larger CFR than females in the early stage but a lower CFR in the late stage. Moreover, the overall estimated median of SOR and SOD were 17.0 and 9.5 days, respectively, but the SOR and SOD were both shortened with the progression of the epidemic. Age, symptoms such as fever, and breathing issues were significant risk factors associated with longer SOR and shorter SOD. However, we found that vaccination, especially two doses, significantly prolonged SOD.

We found that were more male than female COVID-19 cases in KP, but females had a larger CFR than males. Moreover, males were associated with longer SOD (OR = 1.13, *p* < 0.001). However, a study in the district of Attock, Pakistan, found that males aged 40–60 had a significantly higher mortality rate than females [7]. Similarly, studies in India have shown that older people and younger males have a higher risk of death [31]. Studies based on cases in China, Italy, and the UK have also reported higher mortality rates for males [32,33,34]. These findings reflect differences in gender and mortality across geographic regions, and the contradictory results call for further research to identify the relationship between females in KP and increased CFR in COVID-19 cases.

Previous studies have assessed the prevalence of comorbidities in old patients [35,36]. We also found that cases over 60 years and with cardiovascular diseases, chronic lung disease, and diabetes had shorter SOD. Paying more attention to these individuals could reduce the mortality rate. Additionally, cases who followed the epidemic prevention policies, such as home quarantine, were less likely to enter ICU (*p* < 0.001), indicating that epidemic prevention policies tend to be associated with ICU rate. Persisting with epidemic prevention policies might still be the most effective way to control the ICU rate of COVID-19 cases, which is meaningful for regions with large population, especially those lacking medical resources. Our analysis also indicated that ICU case with ventilator was more likely to take a shorter time to die. We noticed that once admitted to ICU, the probability of dying was higher within a week, so a strong and effective treatment or intervention in the first week might be the means to prolong the life of these patients.

The present study also had several limitations. First, we failed to assess the effect of vaccine dose on the length of hospitalization because we do not yet have information on the date of admission to hospital. Second, there was missing information about the vaccination status, but we simply ignored the missing observations, which might introduce bias in the results. Third, cases of Alpha or Delta variants were determined for convenience based on news reports of the Alpha and Delta periods rather than using their sequencing results as the sequencing data were not easy to access.

The outbreak is not quite over, and the current goal remains to reduce the number of COVID-19 cases. We encourage local residents to actively respond to the epidemic prevention and control policies and get vaccinated as soon as possible if there are no special circumstances.

## Figures and Tables

**Figure 1 ijerph-19-11531-f001:**
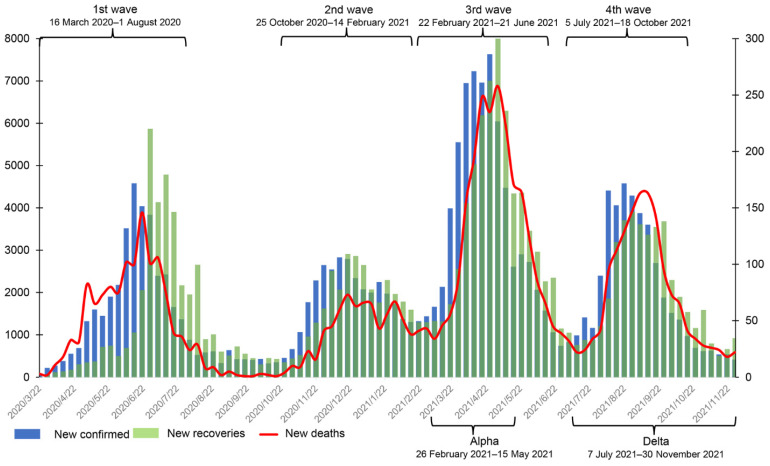
The weekly number of new confirmed cases, deaths, and recoveries from 16 March 2020 to 30 November 2021 in Khyber Pakhtunkhwa, Pakistan. The data were obtained from the official website of the Pakistan government (https://covid.gov.pk/stats/kpk (accessed on 16 August 2022)).

**Figure 2 ijerph-19-11531-f002:**
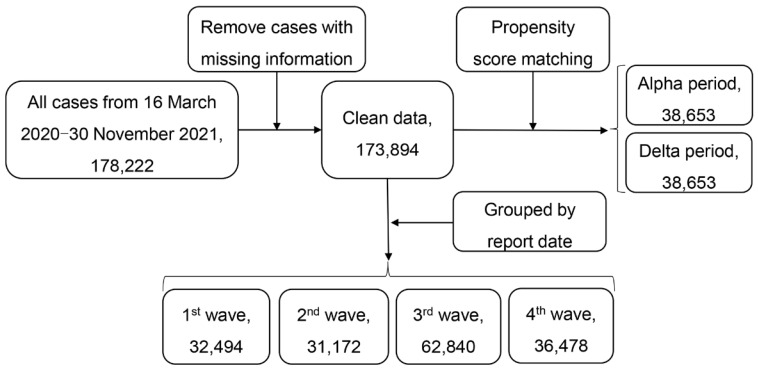
Flowchart of data selection.

**Figure 3 ijerph-19-11531-f003:**
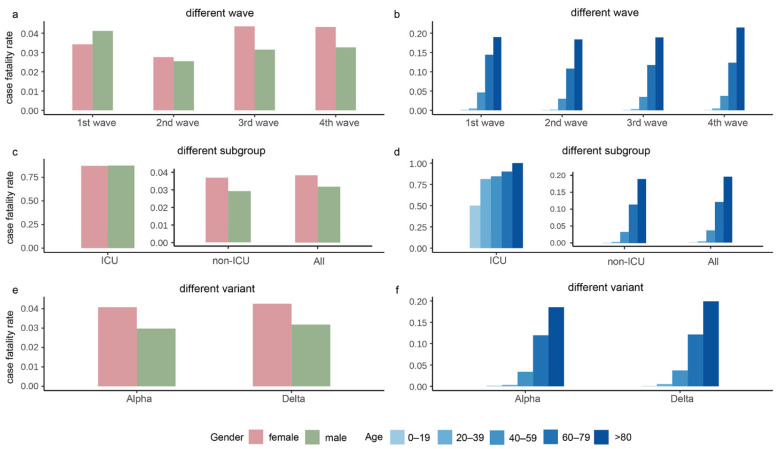
The case fatality rates in the four waves of the pandemic, intensive care unit (ICU)/non-ICU subgroups, and the Alpha and Delta variants. Panels (**a**,**b**) show the case fatality rates grouped by gender and age, respectively, in the four waves. Panels (**c**,**d**) display the case fatality rates grouped by gender and age, respectively, in ICU, non-ICU, and all cases. Panels (**e**,**f**) exhibit the case fatality rates grouped by gender and age, respectively, in the period of Alpha and Delta variants.

**Figure 4 ijerph-19-11531-f004:**
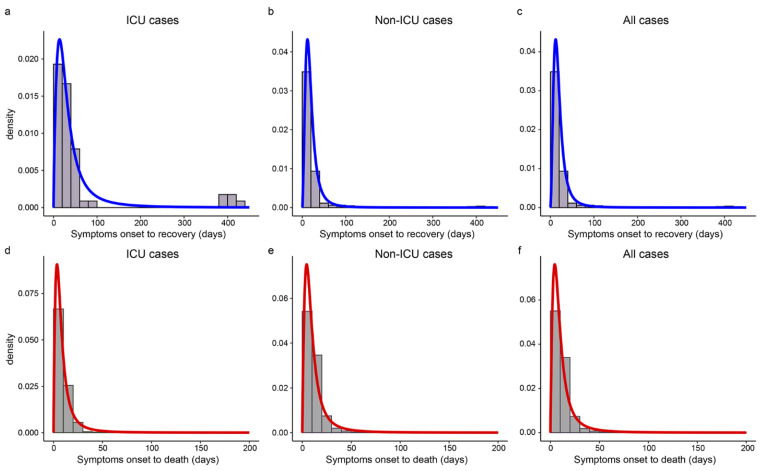
The estimated distributions of SOR and SOD in different subgroups. Panels (**a**–**c**) show the histograms and the estimated density curves of SOR in ICU, non-ICU, and all cases, respectively. Panels (**d**–**f**) display the histograms and the estimated distributions of SOD in ICU, non-ICU, and all cases, respectively.

**Figure 5 ijerph-19-11531-f005:**
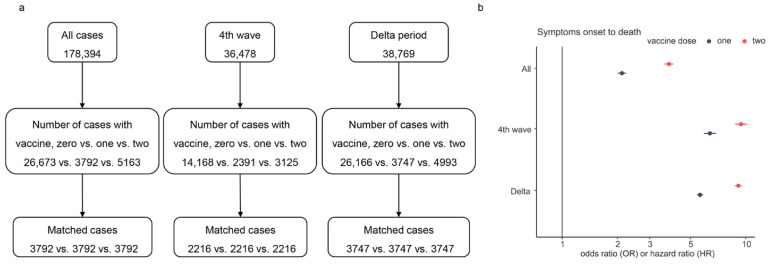
Multiple PSM processes and the odds/hazard ratio of vaccine doses on SOD. Panel (**a**) is the flowchart of multiple PSM for vaccine doses, and panel (**b**) shows the estimated OR/HR values of vaccine doses, where only the ratio of the fourth wave is HR and others are OR.

**Figure 6 ijerph-19-11531-f006:**
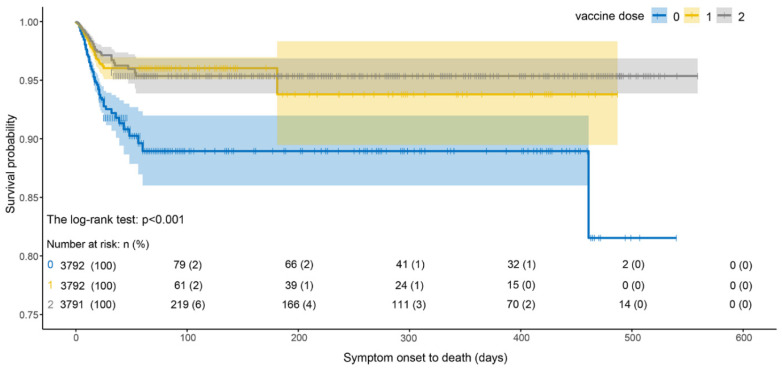
Survival curves of SOD estimated by Kaplan–Meier estimator under different vaccine doses.

**Table 1 ijerph-19-11531-t001:** Demographic characteristics of all SARS-CoV-2 cases and matched cases during the Alpha and Delta periods.

Clinical Features	ICU (%)	Non-ICU (%)	Overall (%)	*p*-Value	Alpha (%)	Delta (%)	*p*-Value
	n = 458	n = 173,436	n = 173,894		n = 38,653	n = 38,653	
Gender				<0.001			0.93
Female	116 (25.3)	64,101 (37.0)	64,217 (36.9)		16,507 (42.7)	16,520 (42.7)	
Male	342 (74.7)	109,335 (63.0)	109,677 (63.1)		22,146 (57.3)	22,133 (57.3)	
Age, years				<0.001			0.999
Median (IQR)	60 (50, 67)	36 (26, 52)	36 (26, 52)		39 (27, 55)	36 (25, 52)	
0–19	4 (0.9)	22,616 (13.0)	22,620 (13.0)		5234 (13.5)	5247 (13.6)	
20–39	32 (7.0)	71,754 (41.4)	71,786 (41.3)		16,234 (42.0)	16,234 (42.0)	
40–59	186 (40.6)	49,468 (28.5)	49,654 (28.6)		10,032 (26.0)	10,032 (26.0)	
60–79	212 (46.3)	26,081 (15.0)	26,293 (15.1)		6101 (15.8)	6101 (15.8)	
>80	24 (5.2)	3517 (2.0)	3541 (2.0)		1052 (2.7)	1039 (2.7)	
Vaccine doses				-			<0.001
0	0(0.0)	26,673 (15.4)	26,673 (15.3)		70 (0.2)	26,065 (67.4)	
1	0(0.0)	3792 (2.2)	3792 (2.2)		13 (0.0)	3739 (9.7)	
2	0(0.0)	5163 (3.0)	5163 (3.0)		35 (0.1)	4986 (10.0)	
Patient status				<0.001			<0.001
Active	0 (0.0)	742 (0.4)	742 (0.4)		9 (0.0)	700 (1.8)	
Died	400 (87.3)	5540 (3.2)	5940 (3.4)		1332 (3.5)	1408 (3.7)	
Recovered	58 (12.7)	167,154 (96.4)	167,212 (96.2)		37,312 (96.5)	36,545 (94.5)	
Symptom							
Fever	374 (81.7)	34,947 (20.1)	35,321 (20.3)	<0.001	7402 (19.1)	8249 (21.3)	<0.001
Sore Throat	68 (14.8)	13,999 (8.1)	14,067 (8.1)	<0.001	2983 (7.7)	3533 (9.1)	<0.001
Cough	371 (81.0)	29,608 (17.1)	29,979 (17.2)	<0.001	6421 (16.6)	6324 (16.4)	0.352
Diarrhea	17 (3.7)	1966 (1.1)	1983 (1.1)	<0.001	339 (0.9)	477 (1.2)	<0.001
Breathing issue	337 (73.6)	13,436 (7.7)	13,773 (7.9)	<0.001	3086 (8.0)	2923 (7.6)	0.030
Headache	7 (1.5)	5611 (3.2)	5618 (3.2)	0.054	1474 (3.8)	1757 (4.5)	<0.001
Basic diseases							
Cardiovascular diseases including hypertension	245 (53.5)	3748 (2.2)	3993 (2.3)	<0.001	648 (1.7)	864 (2.2)	<0.001
chronic lung disease	48 (10.5)	641 (0.4)	689 (0.4)	<0.001	35 (0.1)	354 (0.9)	<0.001
Diabetes	194 (42.4)	2840 (1.6)	3034 (1.7)	<0.001	467 (1.2)	703 (1.8)	<0.001
Pregnancy	0 (0.0)	244 (0.4)	244 (0.4)	1.0	10 (0.1)	162 (1.0)	<0.001
Obesity	2 (0.4)	380 (0.2)	382 (0.2)	0.266	10 (0.0)	334 (0.9)	<0.001
Home quarantine				<0.001			<0.001
Yes	1 (0.2)	102,251 (59.0)	102,252 (58.8)		24,666 (63.8)	19,696 (51.0)	
No	457 (99.8)	71,185 (41.0)	71,642 (41.2)		13,699 (36.2)	18,669 (49.0)	
Put-on Ventilator				<0.001			-
Yes	363 (79.3)	248 (0.1)	611 (0.4)		0 (0.0)	0 (0.0)	
No	95 (20.7)	173,188 (99.9)	173,283 (99.7)		38,653 (100.0)	38,653 (100.0)	

**Table 2 ijerph-19-11531-t002:** The estimated medians of SOR and SOD based on the selected models.

Scenarios		The Selected Model with the Smallest AIC	Median (95%CI)		95th Percentile (95%CI)	
ICU						
	SOR, days	Log-logistic	27.6	(21.0, 36.4)	140.9	(75.0, 247.0)
	SOD, days	Log-logistic	6.9	(6.3, 7.7)	34.5	(27.1, 43.6)
Non-ICU						
	SOR, days	Log-logistic	17.0	(16.9, 17.1)	55.6	(55.1, 56.1)
	SOD, days	Log-logistic	8.4	(8.2, 8.7)	40.2	(37.7, 42.8)
All						
	SOR, days	Log-logistic	17.0	(16.9, 17.1)	55.6	(55.1, 56.1)
	SOD, days	Log-logistic	9.5	(9.2, 9.7)	41.7	(39.4, 44.4)
Alpha						
	SOR, days	Log-logistic	16.0	(15.9, 16.1)	37.6	(37.1, 38.1)
	SOD, days	Gamma	9.7	(9.1, 10.3)	32.6	(30.3, 34.7)
Delta						
	SOR, days	Log-logistic	15.0	(14.9, 15.1)	33.2	(32.8, 33.6)
	SOD, days	Gamma	9.2	(8.7, 9.7)	28.5	(26.7, 30.2)

**Table 3 ijerph-19-11531-t003:** The estimated medians of SOR and SOD for cases with vaccination information.

	Vaccine Doses	The Selected Model with the Smallest AIC	Median (95%CI)		95th Percentile (95%CI)	
**SOR**						
	0	Log-logistic	15.0	(14.9, 15.1)	34.4	(33.9, 35.0)
	1	Log-logistic	15.9	(15.5, 16.2)	39.2	(37.5, 41.1)
	2	Log-logistic	16.5	(16.1, 16.8)	52.4	(49.8, 55.3)
**SOD**						
	0	Log-logistic	9.1	(8.6, 9.6)	39.0	(34.1, 44.3)
	1	Log-logistic	9.4	(7.8, 11.1)	32.7	(22.6, 45.5)
	2	Gamma	11.2	(9.4, 13.3)	30.9	(24.9, 37.0)

**Table 4 ijerph-19-11531-t004:** The median estimates of SOR and SOD in the four waves.

	Wave	The Selected Model with the Smallest AIC	Median (95%CI)		95th Percentile (95%CI)	
**SOR**						
	First wave	Log-logistic	24.7	(24.4, 25.1)	145.9	(141.3, 150.3)
	Second wave	Log-logistic	16.1	(16.0, 16.3)	53.9	(52.8, 55.1)
	Third wave	Log-logistic	16.2	(16.1, 16.3)	42.2	(41.7, 42.7)
	Fourth wave	Log-logistic	15.0	(14.9, 15.1)	32.8	(32.4, 33.2)
**SOD**						
	First wave	Log-logistic	7.6	(7.2, 8.1)	40.5	(35.0, 46.4)
	Second wave	Log-logistic	9.1	(8.5, 9.8)	44.1	(37.6, 52.8)
	Third wave	Gamma	8.9	(8.4, 9.3)	29.6	(28.0, 31.2)
	Fourth wave	Gamma	8.6	(8.1, 9.0)	27.2	(25.6, 28.9)

**Table 5 ijerph-19-11531-t005:** Factors associated with the length of SOR and SOD in all confirmed cases.

Characteristic	Factors Associated with SOR			Factors Associated with SOD		
	OR	95% CI	*p*-Value *	OR	95% CI	*p*-Value *
Gender						
Female	1.00	—	—	1.00	—	—
Male	1.02	(1.02, 1.03)	<0.001	1.13	(1.05, 1.21)	<0.001
Age group, years						
0–19	1.00	—	—	1.00	—	—
20–39	1.04	(1.03, 1.05)	<0.001	0.41	(0.26, 0.66)	<0.001
40–59	1.05	(1.04, 1.06)	<0.001	0.05	(0.03, 0.08)	<0.001
60–79	1.06	(1.05, 1.07)	<0.001	0.02	(0.01, 0.03)	<0.001
80–120	1.07	(1.04, 1.09)	<0.001	0.01	(0.01, 0.02)	<0.001
Symptom						
Fever	1.13	(1.12, 1.14)	<0.001	0.66	(0.59, 0.73)	<0.001
Sore throat	1.03	(1.01, 1.04)	<0.001	1.13	(1.02, 1.25)	0.026
Cough	1.09	(1.08, 1.11)	<0.001	0.94	(0.84, 1.05)	0.267
Diarrhea	1.06	(1.03, 1.09)	<0.001	1.13	(0.90, 1.40)	0.273
Breathing issue	1.06	(1.04, 1.07)	<0.001	0.19	(0.17, 0.21)	<0.001
Headache	0.98	(0.96, 0.99)	0.011	1.32	(1.15, 1.52)	<0.001
Basic disease						
Cardiovascular diseases including hypertension	1.02	(0.99, 1.05)	0.140	0.61	(0.55, 0.69)	<0.001
Chronic lung disease	1.00	(0.94, 1.11)	0.982	0.53	(0.39, 0.70)	<0.001
Diabetes	1.02	(0.99, 1.05)	0.375	0.73	(0.64, 0.82)	<0.001
Obesity	0.95	(0.95, 1.22)	0.384	4.10	(2.08, 8.08)	<0.001
Home quarantine						
No	1.00	—	—	1.00	—	—
Yes	1.06	(1.05, 1.07)	<0.001	9.49	(10.38, 13.07)	<0.001
ICU admission						
No	1.00	—	—	1.00	—	—
Yes	1.00	(0.84, 1.19)	0.982	0.55	(0.42, 0.73)	<0.001
Put on ventilator						
No	1.00	—	—	1.00	—	—
Yes	3.82	(2.80, 5.21)	<0.001	0.06	(0.04, 0.07)	<0.001
Date of diagnosis						
16 March 2020–18 September 2020	1.00	—	—	1.00	—	—
19 September 2020–14 February 2021	0.73	(0.73, 0.74)	<0.001	0.49	(0.43, 0.55)	<0.001
15 February 2021–25 June 2021	0.74	(0.73, 0.74)	<0.001	0.31	(0.28, 0.35)	<0.001
26 June 2021–30 November 2021	0.69	(0.68, 0.70)	<0.001	0.27	(0.24, 0.30)	<0.001

NOTE: *p*-value * indicates the adjusted *p*-value using the Benjamini–Hochberg method [30].

## Data Availability

All data are available upon personal request after approval from the concerned office.

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
