# Peer review of "Two-Dose Vaccination Significantly Prolongs the Duration from Symptom Onset to Death: A Retrospective Study Based on 173,894 SARS-CoV-2 Cases in Khyber Pakhtunkhwa, Pakistan"

_ijerph, 2022, doi:10.3390/ijerph191811531_

Round 1

Reviewer 1 Report

It is my pleasure to review this work. Using data from the office of the Directorate General Health Services (KP), Pakistan, the authors found that the infected vaccinated individuals had longer SOD than unvaccinated individuals, especially for the cases that received two vaccine doses. I have a few suggestions that may improve the manuscript.

1. Line 112, “Since there were a large number of observations with the recorded dates of symptoms onset and death occurred on the same day, which caused the observed duration (unit: days) to be zero”, how many people developed symptoms and died on the same day? Please provide the specific number.

2. Line 129, “We employed accelerated failure time (AFT) models to evaluate the factors associated with SOR and SOD and quantify the effects using the odds ratio (OR) and their 95% CIs”, why use OR instead of hazard ratio (HR) for the effect measure here? Is any reference available for effect measure?

3. Please check and use full names before introducing abbreviations in the abstract and the main text.

4. The full name of the abbrevations in the figures should be listed in the footnote.

5. Could the authors provide AIC values for the four parameter models?

6. Figure 4, does the abscissa represent the time to recovery or death from symptom onset? If so, please add some explanations and mark it clearly.

7. Figure 4, I suggest that the histogram and probability density curve could use different colors for clarity.

8. Table 3, the authors used different models in describing the median values for receiving different doses of the vaccine, which may have caused some confusion.

9. It is recommended to redraw the figure 6 to more clearly distinguish the survival curves of SOD under different vaccine-dose levels.

10. Line 253, “From the descriptive analysis, we see that 69.9% of cases were between the ages of 20-59”, I suggest putting descriptive sentences from the discussion in the results.

Author Response

Thank you very much for the careful reading and thoughtful comments that guide us to improve the manuscript. We have carefully incorporated your comments in this revision.

Our point-by-point responses to your comments are given below, with your original comments copied in bold Italics for your convenience.

  1. Line 112, “Since there were a large number of observations with the recorded dates of symptoms onset and death occurred on the same day, which caused the observed duration (unit: days) to be zero”, how many people developed symptoms and died on the same day? Please provide the specific number.

Response: Thank you for your comment. In the data, there are 287 cases who developed symptoms and died on the same days among 5,835 deaths and 727 cases who developed symptoms and recovered on the same days among 166,860 recoveries. Considering that there exists real possibility that the developed symptoms and death happened on the same day and the 1,014 cases also carry useful information, so we decided to keep them in the analysis. We have added the number in the revision, please see the second paragragh on page 5.

  1. Line 129, “We employed accelerated failure time (AFT) models to evaluate the factors associated with SOR and SOD and quantify the effects using the odds ratio (OR) and their 95% CIs”, why use OR instead of hazard ratio (HR) for the effect measure here? Is any reference available for effect measure?

Response: Thank you for your comment. This is a good question. The AFT model is a general model and can be very flexible as the model error term follows different distributions [1]. Therefor, the explanation of its regression coefficients is also flexible and varies with the type of the error distribution. In specific, if the error distribution is set as the log-logistic distribution, the corresponding effect measure of the coefficients equivalents to the odds ratio (OR); while if the error distribution is set as Weibull, Gamma, or log-normal distribution, the corresponding effect measure equivalents to hazard ratio (HR) [2, 3]. In the revision, considering that AIC selected both the log-logistic model and the Gamma model, so we use odds ratio or hazard ratio according to the specific situation for effect measure.

Reference for the effect measure:

[1] John D. Kalbfleisch, R.L.P. The statistical analysis of failure time data, 2nd ed.; John Wiley, Hoboken, New Jersey, 2002; pp. 235-241.

[2] Qi, J. Comparison of Proportional Hazards and Accelerated Failure Time Models. University of SASKATCHEWAN, 2009.

[3] Shankar Prasad Khanal, V.S., Subrat K. Acharya. Accelerated Failure Time Models: An Application in the Survival of Acute Liver Failure Patients in India. International Journal of Science and Research 2014, 3.

  1. Please check and use full names before introducing abbreviations in the abstract and the main text.

Response: Thank you. We have checked all abbreviations and added their full names in the revision.

  1. The full name of the abbreviations in the figures should be listed in the footnote.

Response: We have removed the abbreviations in all figures for the convenience of reading.

  1. Could the authors provide AIC values for the four parameter models?

Response: Thank you for the comment. Following your suggestion, we presented the AIC values of all parametric models under different scenarios in the supplementary material. Please refer to supplementary materials for details.

  1. Figure 4, does the abscissa represent the time to recovery or death from symptom onset? If so, please add some explanations and mark it clearly.

Response: Thanks for pointing out this detail. The answer is yes. In the revision, we have marked the x-axis label of Figure 4 using full names. Please refer to Figure 4 for details in Line 203.

  1. Figure 4, I suggest that the histogram and probability density curve could use different colors for clarity.

Response: Thank you. It is a great suggestion. We have revised it following your suggestion.

  1. Table 3, the authors used different models in describing the median values for receiving different doses of the vaccine, which may have caused some confusion.

Response: Thanks for your comment. Because the distribution of the duration associated with different dose level of vaccine might be distinct, we did not fit them using one single model. Instead, we assumed no specific model for them. We allowed the fitting model across the Weibull, Gamma, Log-normal, and Log-logistic types, and then used the AIC value to choose the best fitting model.

  1. It is recommended to redraw the figure 6 to more clearly distinguish the survival curves of SOD under different vaccine-dose levels.

Response: Thank you for the suggestion. We have redrawn the figure. Please refer to Figure 6 for details in Line 267.

  1. Line 253, “From the descriptive analysis, we see that 69.9% of cases were between the ages of 20-59”, I suggest putting descriptive sentences from the discussion in the results.

Response: This is a great suggestion. We have moved this sentence to Section 3.1 in lines 158-159,  and rephrased the first sentence of Discussion.

Reviewer 2 Report

The text in most figures should be larger and sharper - Figures 1, 3, 4, 5, and 6.

This article includes multiple statistical calculations but no mention of correction for multiple testing.  Please include text indicating that results are not corrected for multiple testing or alternatively add the multiple testing adjusted P-values in addition to the uncorrected P-values.

Figure 1 appears to have two shades of green bars in the graph, but only one shade is included in the figure legend.  What does the lighter green color represent?

The article does not indicate what COVID-19 vaccines are approved for distribution in Pakistan relevant to the time period of this study.  Please include this information as relevant background.

Line 154-155 - This reviewer does not understand what is intended by the description of "home isolation awareness of ICU cases tended to be poor".

Please confirm that the Non-ICU number for vaccine doses represents significant missing information (roughly 79.4%) - 15.4% + 2.2% + 3.0% only totals to 20.6%

Note that the left side of Figure 5 bottom has been clipped.

Author Response

We thank you for your insightful comments. All these comments have greatly helped us to improve the quality of our manuscript. Our point-by-point responses to your comments are given below, with your original comments copied in bold Italics for your convenience.

  1. The text in most figures should be larger and sharper - Figures 1, 3, 4, 5, and 6.

Response: Thank you for your comment. Following your suggestion, we have adjusted all figures and enlarged the font size of text in them. Please refer to the figures for details.

  1. This article includes multiple statistical calculations but no mention of correction for multiple testing. Please include text indicating that results are not corrected for multiple testing or alternatively add the multiple testing adjusted P-values in addition to the uncorrected P-values.

Response: Thank you for pointing this out. We indeed used the adjusted p-values. We have clarified it in the revised manuscript.

  1. Figure 1 appears to have two shades of green bars in the graph, but only one shade is included in the figure legend. What does the lighter green color represent?

Response: Thank you for the comment. First, Figure 1 is not a stacking histogram. In Figure 1, the blue bar presents the weekly number of new confirmed, and the green bar presents the weekly number of recoveries. Because the blue bar and the green bar have overlap and share the same axes, the overlap region shows lightgreen color. A similar kind of figure can be referred to the Figure 3(B) in an article by Chinese Center for Disease Control and Prevention. (http://dx.doi.org/10.3760/cma.j.issn.0254-6450.2020. 02.003)

  1. The article does not indicate what COVID-19 vaccines are approved for distribution in Pakistan relevant to the time period of this study. Please include this information as relevant background.

Response: This is a great suggestion. Following your suggestion, we added new literature in the Introduction, and fully presented the types of COVID-19 vaccines approved for distribution in Pakistan during Feb 2, 2021-Nov 30, 2021. We also added some information on the vaccination population. Please refer to Lines 68-72.

“From Feb 2, 2021 to Nov 30, 2021, types of COVID-19 vaccines approved for distribution in Pakistan included Cansino, SinoPharm / Beijing, Sputnik V, followed by Sinovac, Oxford / Astrzeneca, Covaxin, Modema, and other vaccine brands such as Pfizer / Biontech [10]. The government of Pakistan was providing the CanSino, Sinopharm, SinoVac, and AstraZeneca vaccines free of cost. Health care workers were the first to be vaccinated [11]. After March 16, 2021, vaccination for different ages began [12]. ”

Reference

[10] Coronavirus (COVID-19) Vaccinations. Available online: https://ourworldindata.org/ (accessed on 2 September 2022).

[11] Khan, S.; Uddin, A.; Imran, M.; Ali, Y.; Khan, S.; Salman Khan, M.; Trutter, B.; Asfandiyar, M.; Iqbal, Z. COVID-19 Vaccine Acceptance and Hesitancy among Health Care Workers (HCWs) In Two Major Urban Centers in Khyber-Pakhtunkhwa, Pakistan. Asia Pacific Journal of Public Health 2022, 34, 580-582, doi:10.1177/10105395221083382.

[12] Siddiqui, A.; Ahmed, A.; Tanveer, M.; Saqlain, M.; Kow, C.S.; Hasan, S.S. An overview of procurement, pricing, and uptake of COVID-19 vaccines in Pakistan. Vaccine 2021, 39, 5251-5253, doi:10.1016/j.vaccine.2021.07.072.

  1. Line 154-155 - This reviewer does not understand what is intended by the description of "home isolation awareness of ICU cases tended to be poor".

Response: Thank you for your comments. We aimed to show that the epidemic prevention policy such as home isolation tends to be associated with the severity of the patient's illness condition. Cases who follow the epidemic prevention policy are less likely to enter the ICU once infected. We have revised the presentation in the revision.

  1. Please confirm that the Non-ICU number for vaccine doses represents significant missing information (roughly 79.4%) - 15.4% + 2.2% + 3.0% only totals to 20.6%.

Response: Thank you for the careful reading. Yes, missing information on vaccine doses in non-ICU cases exists. Some cases may have been vaccinated, while the information was not recorded. In this study, to investigate the effect of dose levels on SOD, we only considered the cases with records of vaccination. In total, there are 35,649 cases having recorded vaccine doses.

  1. Note that the left side of Figure 5 bottom has been clipped.

Response: Thank you. We have revised Figure 5. Please refer to the main text for details in Lines 262-265.
